# SocialGPT: Prompting LLMs for Social Relation Reasoning via Greedy Segment Optimization

**Wanhua Li**[*,1] **Zibin Meng**[*,1,2] **Jiawei Zhou**[3] **Donglai Wei**[4] **Chuang Gan**[5,6] **Hanspeter Pfister**[1]

[1]Harvard University   [2]Tsinghua University   [3]Stony Brook University
[4]Boston College   [5]MIT-IBM Watson AI Lab   [6]UMass Amherst

## Abstract

Social relation reasoning aims to identify relation categories such as friends, spouses, and colleagues from images. While current methods adopt the paradigm of training a dedicated network end-to-end using labeled image data, they are limited in terms of generalizability and interpretability. To address these issues, we first present a simple yet well-crafted framework named SocialGPT, which combines the perception capability of Vision Foundation Models (VFMs) and the reasoning capability of Large Language Models (LLMs) within a modular framework, providing a strong baseline for social relation recognition. Specifically, we instruct VFMs to translate image content into a textual social story, and then utilize LLMs for text-based reasoning. SocialGPT introduces systematic design principles to adapt VFMs and LLMs separately and bridge their gaps. Without additional model training, it achieves competitive zero-shot results on two databases while offering interpretable answers, as LLMs can generate language-based explanations for the decisions. The manual prompt design process for LLMs at the reasoning phase is tedious and an automated prompt optimization method is desired. As we essentially convert a visual classification task into a generative task of LLMs, automatic prompt optimization encounters a unique long prompt optimization issue. To address this issue, we further propose the Greedy Segment Prompt Optimization (GSPO), which performs a greedy search by utilizing gradient information at the segment level. Experimental results show that GSPO significantly improves performance, and our method also generalizes to different image styles. The code is available at `https://github.com/Mengzibin/SocialGPT`.

## 1 Introduction

Social relationships are of paramount importance in our lives, as they significantly impact our emotional, psychological, and physical well-being. Social relationship recognition aims to categorize the relationships such as friends, colleagues, band members, and so on, that exist between individuals given an input image and the bounding boxes of the two persons of interest [1]. In recent years, social relationship recognition has garnered significant attention [1–4] due to its wide range of applications, including product recommendation [5], autonomous systems [6], and more.

Over the past decade, the field of computer vision has witnessed tremendous success [7–12] in the end-to-end learning framework, which trains a dedicated neural network end-to-end on a customized dataset. Research in social relationship recognition has also followed a similar trajectory [1, 13, 2]. As social relationship reasoning represents a cognitive function that operates at a higher level than visual perception, many methods [6, 3] incorporate rich prior knowledge of social relations into the models. For example, GRM [6] integrated a knowledge graph into its model to leverage the

---

*Equal contribution.

38th Conference on Neural Information Processing Systems (NeurIPS 2024).

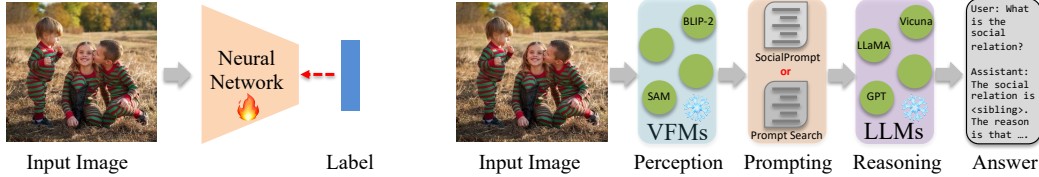

| (a) End-to-end Learning-Based Framework | (b) Modular Framework with Foundation Models |

Figure 1: (a) End-to-end learning-based framework for social relation reasoning. A dedicated neural network is trained end-to-end with full training data. (b) We propose a modular framework with foundation models for social relation reasoning. Our proposed SocialGPT first employs VFMs to extract visual information into textual format, and then perform text-based reasoning with LLMs, using either our manually designed SocialPrompt or optimized prompts.

information of contextual objects. GR$^2$N [3] and TRGAT [14] exploit the logical constraints among multiple social relationships within the same scene. While these methods have achieved notable results, they are limited in terms of generalization and interpretability. In other words, we cannot trust that the trained models can generalize to arbitrary scenarios, and these models fail to provide the reasons and explanations for their decisions.

In this paper, we first present a modular framework with foundation models for social relation reasoning. Recently, we have witnessed the significant success of foundational models [15]. Many Vision Foundation Models (VFMs) can accurately perform basic visual perception tasks such as identifying "what" and "where" in images [16–19]. On the other hand, the emergence of Large Language Models (LLMs) demonstrates strong reasoning capabilities [20–23]. Therefore, we present a framework that follows the "perceive with VFMs, reason with LLMs" paradigm. This framework first employs VFMs to convert images into textual data, and subsequently leverages the textual reasoning capabilities of LLMs for relation prediction. In this process, VFMs process visual signals into fundamental facts, and then LLMs analyze these facts to make explainable inferences.

Our framework performs visual reasoning for **Social** relationship recognition using **GPT**-style LLMs, coined SocialGPT. SocialGPT introduces systematic design principles to guide and adapt VFMs and LLMs for social relationship reasoning. Specifically, in the perception phase, we extract both comprehensive and domain-specific visual information with VFMs, which is further fused into a coherent textual social story with symbol-based object reference and is easily readable. In the reasoning phase, we utilize a structured social relation reasoning prompt, named SocialPrompt, composed of different segments for "system, expectation, context, and guidance" to better instruct LLMs. With the proposed systematic design principles, our SocialGPT provides a strong baseline and achieves highly competitive zero-shot results, compared to the state-of-the-art methods that undergo end-to-end training on full training datasets.

Lastly, we observed that LLMs exhibit high sensitivity to prompts during the reasoning process, but the manual prompt design is a time-consuming and labor-intensive task [24, 25]. We propose the Greedy Segment Prompt Optimization (GSPO) algorithm for automatic prompt tuning. As we convert a visual classification task as a generative task of LLMs, automatic prompt tuning for SocialPrompt encounters the long prompt optimization issue. Our proposed GSPO addresses these issues by utilizing gradient information at the segment level for greedy search. Experiments demonstrate that GSPO significantly improves the performance of LLMs. Figure 1 visualizes our paradigm. To summarize, we make the following contributions: 1). We present a simple modular framework with foundation models for social relation reasoning, which provides a strong baseline as the first zero-shot social relation recognition method. 2). To address the long prompt optimization issue associated with visual reasoning tasks, we further propose the Greedy Segment Prompt Optimization, which performs a greedy search on the segment level with gradient guidance. 3). Experiments demonstrate that our method attains very competitive and explainable zero-shot results without additional model training. With GSPO, our method significantly outperforms the state-of-the-art methods.

## 2 Related Work

**Foundation Models.** Recently, we have witnessed the tremendous success of foundational models [19, 26–29]. Foundation models are typically trained on massive data, possess a large number

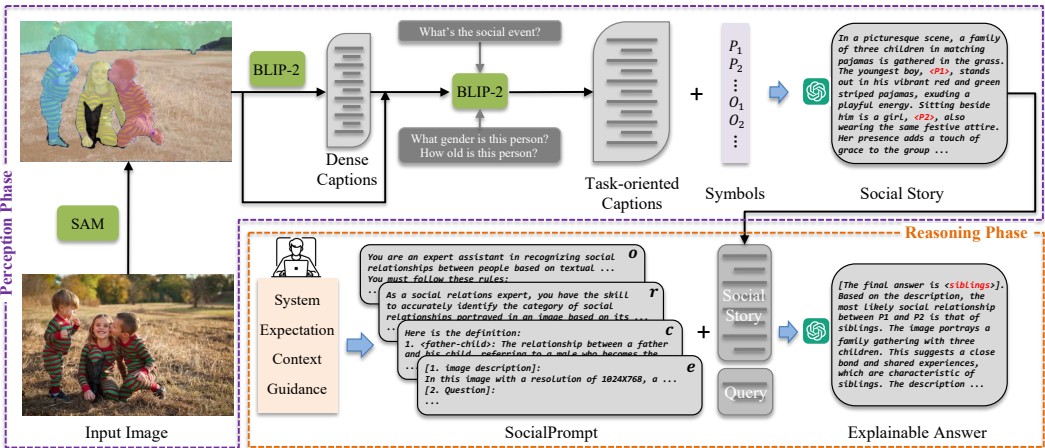

Figure 2: The framework of SocialGPT, which follows the "perception with VFMs, reasoning with LLMs" paradigm. SocialGPT converts an image into a *social story* in the perception phase, and then employs LLMs to generate explainable answers in the reasoning phase with SocialPrompt.

of model parameters, and exhibit excellent performance along with strong generalization capabilities [15]. The emergence of LLMs [15, 27, 30, 31] has significantly reshaped the field of Natural Language Processing (NLP). ChatGPT and GPT-4 [27], developed by OpenAI, are among the most famous LLMs. GPT-4, in particular, demonstrates a strikingly close-to-human-level intelligence [32]. Meanwhile, many open-source LLMs like Vicuna [29], LLaMa [33], and LLaMa-2 [34] have been developed, and have achieved outstanding performance across various NLP tasks. On the other hand, VFMs [19, 26, 35–38] have also made significant advancements. CLIP [19] connects images and text, enabling zero-shot image classification [39, 40]. BLIP [41] and BLIP-2 [16] demonstrate strong zero-shot image-to-text generation capabilities. SAM [17] offers a foundation model for image segmentation [42]. While foundation model-based frameworks have been proposed for many other tasks including few-shot visual recognition [43–45], visual question answering [46–48], and semantic segmentation [49], our SocialGPT explicitly employs text as the bridge between VFMs and LLMs and then proposes symbol-based referencing to support unambiguous text queries.

**Social Relation Recognition.** Social psychologists have conducted extensive research on social relationships over decades [50, 51], resulting in several different social theories [52, 53]. Sun *et al.* [1] followed Bugental's domain-based theory [52] and annotated the PIPA dataset, which has become one of the most popular benchmarks for social relation recognition. Li *et al.* [13] adopted the relational models theory [53] and contributed the People in Social Context (PISC) dataset. A dual-glance model was further proposed to leverage multiple contextual regions. With the well-established benchmarks, numerous end-to-end methods [54, 3, 14, 2] have been proposed, effectively advancing the field of social relationship recognition. Some methods [54, 6] employed knowledge graphs to exploit scene and global contextual cues. Noticing that there usually are multiple social relations on the same image, Li *et al.* [3] proposed GR$^2$N to jointly infer all relations on an image with graph neural networks. TRGAT [14] further considered higher-order constraints for social relations on an image and achieved better results. These methods adopted the end-to-end learning-based paradigm, whereas we propose a modular framework with foundation models.

## 3 SocialGPT

Social relation recognition takes an image $I$ and two bounding boxes $b_1$ and $b_2$ of two interested individuals as inputs, and requires a model that outputs the social relationship $y$. We first introduce a modular framework with foundation models for social relation recognition in this section, which provides a strong zero-shot baseline. The pipeline is illustrated in Figure 2. On a high level, we first use VFMs to extract visual information at different granularities. The raw information is then fused into a coherent *social story* in textual format, denoted as $S$, which can be best reasoned with LLMs.

## 3.1 Perception with Vision Foundation Models

The perception objective is to extract essential visual information related to social relation reasoning, in order to connect with text-based LLMs for downstream reasoning. One straightforward approach is to utilize existing image captioning foundation models such as BLIP-2 [16] to generate a caption or GPT-4V [55] to generate an image description. However, a single sentence or general-purpose description may overlook crucial details relevant to social relations present in the images.

We construct text-based visual information with VFMs with being both **comprehensive** and **domain-specific** as our guidelines. To achieve this, we resort to the state-of-the-art image segmentation tool, the Segment Anything Model (SAM) [17], and the powerful vision-language foundation model, BLIP-2 [16], for both identifying important details in the image and describing them in language. In particular, we use SAM to segment the image to obtain all different object masks, and then send individual objects by masking out others to BLIP-2 to obtain descriptions of each object. Together with the image-level caption, we formulate the *dense captions* covering all objects in the input image.

The above gives us a comprehensive description of the image details. However, holistic captions of the image and different objects are not tailored to our task of social relation reasoning. To compensate for the lack of domain-specific information, we ask specific questions related to social identities by using the BLIP-2 dialog functionality to extract more specific information depending on object types. Recent research [54, 1] has shown that the age and gender of individuals, as well as the social scene and activity, are important clues. Therefore, we actively inquire BLIP-2 about these clues. Specifically, when dealing with people objects, we inquire about age and gender details. This information is crucial for distinguishing familial relationships within a family unit, such as father-child and grandmother-grandchild relationships. For image-level captions, we explore the social scenario or event depicted in the picture. This approach allows us to generate *task-oriented captions* that are tailored to our social relation recognition objective.

## 3.2 Social Story Generation

One could directly input the dense captions and task-oriented captions along with object axes and dimensions into LLMs for social relation reasoning, but the information is fragmented and objects are described in isolation. On the other hand, LLMs perform the best when working with human-readable natural language and they often struggle with arithmetic reasoning tasks [56–58]. Therefore, we integrate the aforementioned vision information by composing a social story that is complete and coherent. Objects are conveniently **referable** and described in relative relations, and the full story is easily **readable** by both humans and LLMs. This will serve as a crucial bridge from visual perception to textual reasoning, providing a solid foundation for the next step of understanding with LLMs.

We propose *symbol-based referencing* for object referral. Multiple individuals and various social relationships coexist in a single image, and bounding boxes $b_1$ and $b_2$ are provided for specific relation inquiries in supervised learning settings. However, as we now convert the entire image into textual data and rely on LLMs for analysis, effective referral of individual objects becomes a critical question. Based on SAM segmentation masks, we can naturally derive bounding boxes for each object $i$ as $b_i = [x_i, y_i, h_i, w_i]$, where $(x_i, y_i)$ is the center coordinate and $(h_i, w_i)$ are the height and width. While directly using these coordinates for referrals in the social story and question inquiries is precise, they pose extra challenges for readability and numerical reasoning for LLMs. Instead, we assign *symbols* to each object to associate with its coordinates in the original image, textual caption, and task-specific features for our social story generation. We use $P_i$ to refer to people objects, and $O_i$ to refer to other objects. Numerical coordinates will not appear in our social story, and relative positional relations are described with the referral symbols. The symbol-based referring also enables straightforward querying for LLMs. For instance, one can directly inquire LLMs about the social relationship between $P_2$ and $P_3$ with natural language and LLMs will easily identify the queried persons associated with symbols. This provides a clear and concise bridge between the object descriptions and the bounding box-based queries, and a similar method can be adopted for a broader range of applications when text-based reasoning is involved for object referral for visual question answering, robotics, etc.

Finally, based on the list of isolated image and object descriptions after symbol-based referencing, we instruct an LLM to act as an information fusion tool for generating a coherent social story $S$ in a unified paragraph. The social story tells all the information needed about the visual scene for

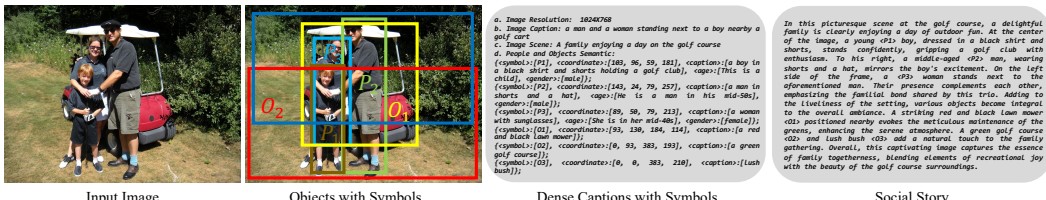



| Input Image | Objects with Symbols | Dense Captions with Symbols | Social Story |

Figure 3: An example of social story generation.



text-based reasoning, which is highly readable and understandable by humans and LLMs with clear symbol references and information consolidation. An example of extracted perceptual information with symbol associations and the generated social story is depicted in Figure 3.

### 3.3 Reasoning with Large Language Models

After obtaining the mapping from image to social story: $I \to S$, we feed both $S$ and bounding box queries $(b_i, b_j)$, converted to textual queries $q$ with referencing symbols $P_i, P_j$, into LLMs to obtain interpretable answers $a$. This is to let LLMs output the map from $(S, q)$ to $a$, which we do by prompting. Since LLM performance is highly sensitive to prompt variations [59, 55], we design our social relation reasoning prompt with four segments, which we name SocialPrompt.

**System.** This is the system prompt provided by many LLMs to steer their behavior. We utilize it to explicitly define several core rules for our task of social reasoning. We denote it as the $o$ segment.

**Expectation.** This is the instruction that we give to the model to set expectations of the anticipated outcomes. This helps avoid vague or unexpected outputs. To do so, we construct a role assignment and task description prompt, denoted as $r$, where we explicitly assign the role of a social relation expert to the LLM and provide a detailed elaboration of the task's input and output.

**Context.** This provides sufficient contextual information to help the LLMs understand the background of the problem. As a classification task, we provide specific definitions for each social relationship category, resulting in the prompt segment denoted as $c$.

**Guidance.** This offers an exemplar to show the LLMs how to respond to a query based on a social story. In-context learning has been proven as an effective means to expand the capabilities of LLMs [60–62]. We manually construct an in-context example prompt, denoted as $e = (S_0, q_0, a_0)$, to better guide LLMs in performing social relationship reasoning in the desired format. Here we also guide the model to generate possible explanations for its prediction. While using more in-context examples may potentially further enhance performance, this is beyond the scope of the paper and is left as future work.

The final SocialPrompt consists of $(o, r, c, e)$, and is concatenated with a testing story-query pair $(S, q)$ at the end for model predictions. Figure 2 shows the structured excerpts of SocialPrompt, and we put the full prompt into the Appendix. Note that we do not use any training samples provided by a dataset and only employ the foundation models. Consequently, SocialGPT is capable of zero-shot social relation reasoning, while maintaining its interpretability and generalizability.

## 4 Greedy Segment Prompt Optimization

Although we have devised well-structured SocialPrompt for social relation reasoning, experiments reveal that different ways of prompt rephrasing and demonstration example variations can significantly impact the LLM reasoning performance. Manually searching for the optimal prompt is time-consuming and labor-intensive, thus automatic prompt tuning is desired. Nevertheless, unlike the prompt optimization methods [63, 64] typically employed in NLP, automatic prompt tuning for SocialPrompt faces two unique challenges: *free-form target* and *long prompt optimization*. As we convert a visual classification task into a generative task for LLMs, the model's output space transitions from discrete numerical representations of one-hot labels to unconstrained textual forms. Defining free-form text objectives for SocialPrompt optimization is not well-explored. Meanwhile, as the social story $S$ is a comprehensive description of the image such as in Figure 3, and task

---

**Algorithm 1** Greedy Segment Prompt Optimization

---

**Input:** Initial segments $\boldsymbol{w}_{1:M}$, training dataset $\mathcal{T}$, iteration number $N$

    Build the candidate set $\mathcal{W}_m$ for each segment $\boldsymbol{w}_m$

    **repeat** $N$ times

        Randomly sample a batch of data $\mathcal{D}$ from $\mathcal{T}$

        **for** $m = 1, \ldots, M$ **do**

            $\mathcal{U}_m := \text{Top-}k(-\sum_{\boldsymbol{z}\in\mathcal{D}} \nabla_{h_{w_m}} \mathcal{L}(\boldsymbol{w}_{1:M}; \boldsymbol{z}))$

                                ▷ *Compute top-k promising segment substitutions*

        **for** $b = 0, 1, \ldots, K * M - 1$ **do**

            $\tilde{\boldsymbol{w}}_{1:M}^{(b)} := \boldsymbol{w}_{1:M}$                                 ▷ *Initialization*

            $\tilde{w}_i^{(b)} := \mathcal{U}_i(\lfloor b/M \rfloor)$, where $i = (b \bmod M) + 1$

                                  ▷ *Select one replacement segment*

        $\boldsymbol{w}_{1:M} := \tilde{\boldsymbol{w}}_{1:M}^{(b^\star)}$, where $b^\star = \arg\min_b \sum_{\boldsymbol{z}\in\mathcal{D}} \mathcal{L}(\tilde{\boldsymbol{w}}_{1:M}^{(b)}, \boldsymbol{z})$     ▷ *Compute best replacement*

**Output:** Optimized segments $\boldsymbol{w}_{1:M}$

---

and full label set definitions could be lengthy, our SocialPrompt tends to be very long. This poses additional challenges for automatic prompt tuning methods. To address these issues, we propose a segment-based optimization algorithm, named Greedy Segment Prompt Optimization (GSPO).

**Tuning Objective.** To automate prompt searching, the first step is to define the optimization objective. Ideally, we aim to find the optimal prompt $\{\boldsymbol{o}^*, \boldsymbol{r}^*, \boldsymbol{c}^*, \boldsymbol{e}^*\}$ that maximize the probability of LLMs generating the correct answer $\boldsymbol{a}$ for any given sample $\boldsymbol{z} = (\boldsymbol{S}, \boldsymbol{q})$. Let's first review the training paradigm commonly used for autoregressive language models [65, 66, 60], which essentially employ the next token prediction task, *i.e.*, learning $p(w_{n+1}|w_{1:n})$, where token $w_{n+1} \in \mathcal{V}$, and $\mathcal{V}$ represents the token vocabulary. Unlike typical classification tasks where only a one-hot formatted category is predicted, our answers are free-form text, consisting of a sequence of numerous tokens. Constructing the ground truth with free-form text for each sample is challenging. This paper proposes instructing LLMs to begin their response with the predicted class category following a pre-defined template. Formally, we assume that the ground truth answer $\boldsymbol{a}$ for sample $\boldsymbol{z}$ takes the following form: $\boldsymbol{a} = [\boldsymbol{a}^0, \boldsymbol{a}^1, \boldsymbol{a}^2, ...]$, where $\boldsymbol{a}^0$ denotes the first sentence of $\boldsymbol{a}$, $\boldsymbol{a}^1$ is the second sentence, and so forth. We specify $\boldsymbol{a}^0$ to have the following fixed format: $\boldsymbol{a}^0 = $ "*The final answer is* $str(\boldsymbol{y})$", where $str(\boldsymbol{y})$ represents the string representation of class label $\boldsymbol{y}$. Then we can define the objective:

$$\mathcal{L}(\boldsymbol{o}, \boldsymbol{r}, \boldsymbol{c}, \boldsymbol{e}; \boldsymbol{z}, \boldsymbol{y}) = -\mathbb{E}_{(\boldsymbol{z}, \boldsymbol{a}^0)} \left[ \log p(\boldsymbol{a}^0|\boldsymbol{o}, \boldsymbol{r}, \boldsymbol{c}, \boldsymbol{e}; \boldsymbol{z}) \right], \tag{1}$$

where the expectation is taken from a collection of training examples, and the probabilities are computed from LLM's next token prediction distributions. Note here the LLM is frozen, and we seek to find the optimal prompt to minimize the above loss. In practice, we employ the same template in our in-context example, making it easy for LLMs to follow a consistent output format. This ensures that the loss primarily stems from LLMs' predictions of tokenized category names rather than category-agnostic sentence formatting. Note that we only construct and supervise the first sentence of the ground truth answer, while the model is free to generate its explanation in the following sentences.

**Long Prompt Optimization.** We optimize over discrete prompt tokens, constrained to a vocabulary $\mathcal{V}$ for each token position associated with the LLM. While some discrete prompt optimization algorithms [67, 25, 67] have been proposed in the NLP field, they typically operate on a limited number of tokens. In contrast, as a visual reasoning task, we require long prompts to adequately convey the dense information and provide detailed context. In fact, the number of tokens in our SocialPrompt may well exceed 2K, and conduct token-level optimization results in a search space of $2000^{|\mathcal{V}|}$, which is beyond the capacities of current optimization methods as $|\mathcal{V}| = 32,000$ for many LLMs [33, 34]. We propose to perform segment-level optimization as a surrogate. Formally, suppose the prompt is $\boldsymbol{w}$ with $M$ segments, denoted as $\boldsymbol{w}_{1:M}$. In our case we can have $M = 4$ and directly map the segments to $\boldsymbol{o}, \boldsymbol{r}, \boldsymbol{c}, \boldsymbol{e}$, respectively. We propose a candidate set $\mathcal{W}_m$ consisting of alternative prompts for each segment, which we use ChatGPT to generate followed by light manual revisions, and the algorithm searches over the combination of different candidates. For the demonstration example segment $\boldsymbol{e}$, we also manually select samples from an existing training set as candidates.

More specifically, inspired by AutoPrompt [25], our optimization algorithm considers all possible single-segment substitutions, thereby selecting the segment candidate that minimizes the loss over

Table 1: The comparison results on the PIPA dataset. ZS stands for Zero-Shot.

| Methods | ZS | Acc (%) |
|---|---|---|
| All attributes + SVM [1] | ✗ | 57.2 |
| Pair CNN [13] | ✗ | 58.0 |
| Dual-Glance [13] | ✗ | 59.6 |
| SRG-GN [54] | ✗ | 53.6 |
| GRM [6] | ✗ | 62.3 |
| MGR [2] | ✗ | 64.4 |
| GR$^2$N [3] | ✗ | 64.3 |
| TRGAT [14] | ✗ | 65.3 |
| SocialGPT (w/ GPT-3.5) | ✔ | 64.1 |
| SocialGPT (w/ Vicuna-13B) | ✔ | **66.7** |

Table 2: Ablations on components of SocialGPT with Vicuna-7B. The results are obtained on the PIPA dataset with a zero-shot setting.

| Methods | Acc (%) |
|---|---|
| SocialGPT | **61.58** |
| - Dense Captions | 52.63 |
| - Task-oriented Captions | 59.89 |
| - Symbol → Object Coordinate | 57.68 |
| - Symbol → Object Caption | 59.83 |
| - Social Story | 45.31 |
| - SocialPrompt Segment {System} | 60.23 |
| - SocialPrompt Segment {Expectation} | 59.19 |
| - SocialPrompt Segment {Context} | 61.18 |
| - SocialPrompt Segment {Guidance} | 43.56 |

a batch of training samples. We replace one segment at a time in a greedy manner. In practice, instead of evaluating all possible candidates, we further reduce the search space by calculating the gradients of the one-hot segment indicators for each segment and selecting the top $K$ most promising candidates for that segment. The gradient is computed as: $\nabla_{h_{w_m}} \mathcal{L}(\boldsymbol{w}_{1:M}) \in \mathbb{R}^{|\mathcal{W}_m|}$, where $h_{w_m}$ represents the one-hot representation of selecting $\boldsymbol{w}_m$ from the set $\mathcal{W}_m$. Then the top $K$ promising substitutions with the largest negative gradient are chosen for evaluation. We repeat this process to acquire $K$ candidates for each segment, and we only replace one segment at a time to obtain $K * M$ new prompts. Then the one with the smallest loss over a batch of training samples is chosen. We iterate this process $N$ times to find the best-performing prompt. The entire search process is shown in Algorithm 1.

## 5 Experiments

### 5.1 Settings

**Data and Evaluation.** We adopt two widely-used benchmarks for social relation reasoning: PIPA [1] and PISC [13]. The PIPA dataset categorizes 16 types of social relationships, including family bonds (like parent-child, grandparent-grandchild), personal connections (friends, loves/spouses), educational and professional interactions (teacher-student, leader-subordinate), and group associations (band, sports team, colleagues). The PISC dataset categorizes social relationships into six types: commercial, couple, family, friends, professional, and no-relation. We follow the standard train/val/test split for both datasets and report the classification accuracy on the test set. Note that the training set is not used for our zero-shot results, but is used for in-context exemplar proposals for our prompt optimization algorithm. For both datasets, we measure classification accuracy as our evaluation metric.

**Implementation Details.** We use two VFM models for visual information extraction – the SAM [17] model for object segmentation, followed by BLIP-2 [41] for dense caption generation. For the social story generation, we employ the GPT-3.5 [55] Turbo model that has empowered ChatGPT. We set the temperature to 0 for greedy decoding to bolster the result's reproducibility. Other generation parameters are otherwise set as default. For subsequent reasoning of social relations based on generated stories, we experiment with both GPT-3.5 and open-source LLMs, including Vicuna-7B/13B [29] and Llama2-7B/13B [34]. All the decoding temperature is set as 0, and we set the maximum context length to 4096 for Vicuna and Llama2 to accommodate our long prompt. For GSPO, we curate $M = 15$ candidates for each of the four segments within the complete prompt and set $K = 3$ for candidate selection for $N = 500$ iterations. One A100 GPU is used for all experiments.

### 5.2 Zero-shot Social Relation Recognition with SocialGPT

**Main Results.** We compare SocialGPT, using either GPT-3.5 or Vicuna-13B, with previous fully supervised methods and present our results in Table 1 and Table 3. Here our method does not

Table 3: The comparison results on the PISC dataset. Previous methods are replicated with open-source code to report the accuracy metric. ZS means Zero-Shot.

| Methods | ZS | Acc (%) |
|---|---|---|
| Pair CNN [13] | ✗ | 46.30 |
| GRM [6] | ✗ | 64.18 |
| GR$^2$N [3] | ✗ | 64.70 |
| SocialGPT (w/ GPT-3.5) | ✔ | 53.43 |
| SocialGPT (w/ Vicuna-13B) | ✔ | **65.12** |

Table 4: Comparison with existing Vision-Language Models on the PIPA dataset, with SocialGPT using Vicuna-13B model.

| Methods | Acc (%) |
|---|---|
| BLIP-2 [41] | 35.84 |
| LLaVA [68] | 45.12 |
| GPT-4V [55] | 59.67 |
| SocialGPT | **66.70** |

undergo the prompt tuning optimization, performing relation reasoning in a zero-shot fashion without utilizing any training examples. On both datasets, Vicuna-13B performs better than GPT-3.5 with our framework. In particular, on PIPA benchmark shown in Table 1, SocialGPT achieves the best accuracy compared with all prior supervised approaches, leading the previous state-of-the-art model TRGAT [14] by 1.4%. The results on the PISC benchmark are shown in Table 3. Most previous methods used mAP (mean Average Precision) as the metric on the PISC dataset, whereas we opted not to employ this metric due to the disparity between our predictions. Unlike previous methods that output per-class confidence scores, our prediction is the textual outputs from LLMs. Therefore, we still adopt the accuracy metric on the PISC dataset. To report the accuracy performance of other methods, we chose the state-of-the-art methods with publicly available code for reproduction and compared their performance. Table 3 shows that our method attains comparable results to the state-of-the-art GR$^2$N model, despite not being trained with any data.

**Comparison with End-to-End VLMs.** Our approach breaks down the social relation reasoning into different phases involving perception tasks with VFMs and reasoning with LLMs, bridged by a coherent textual social story. However, recent advancements in multimodal foundation models (VLMs) provide a straightforward way of reasoning about visual contents, which is simply asking questions about the image to a vision-language model that can respond with an answer directly. We compare SocialGPT with three state-of-the-art end-to-end vision-language foundation models by directly inquiring about social relationships in the image, including BLIP-2 [41], LLaVA [68], and GPT-4V [55], with results shown in Table 4. We see that the method of querying vision-language foundation models, albeit simple, is still lagging behind our approach of SocialGPT with principled designs and modularized VFMs and LLMs. Our well-designed SocialGPT even outperforms the high-performing GPT-4V by 7.03% in accuracy. These results justify the design principles of our framework with comprehensive perception extraction and coherent language reasoning.

**Ablation Study.** We conduct a series of ablation studies to assess the efficacy of various components at different stages of SocialGPT. Table 2 shows the results with Vicuna-7B on the PIPA dataset. The first part of ablation focuses on the social story generation pipeline. As we use SAM to segment the image for visual perception, removing SAM would disable fine-grained object descriptions (dense captions) in the social story, resulting in an accuracy drop of more than 8%. If we do not acquire the task-oriented captions, there is a performance drop of 1.69%. Next, a crucial component of the social story generation in SocialGPT is the utilization of symbols ($P$ for people and $O$ for others) for effective referral of objects. If we do not use the symbols, but instead replace the object referral with either the direct coordinate or the object-specific caption from BLIP-2 in both the social story and the question, we see the performance drops by 3.90% and 1.75%, respectively. Finally, we fuse the multi-aspect visual information into a cohesive social story. If we bypass the fusion and directly utilize the visual annotations from VLMs, we can see there is a significant performance drop of 16.27%. This indicates that a good textual description of comprehensive visual information is necessary to connect LLMs to reason about social scenes presented in images.

We also ablate the SocialPrompt segments in our LLM reasoning phase. We do this by removing each of the segments from the full prompt one at a time, and results are presented in the bottom half of Table 2. We can see that guidance segmentation, which contains a manually constructed demonstration example of how to reason about social relations based on our social story, has the most influence on the model performance. Without it, the accuracy drops by 18.02%. The system

Table 5: Prompt tuning results (accuracy in %) with GSPO.

| Model | PIPA | | | PISC | | |
|-------|------|------|------|------|------|------|
| | SocialGPT | + GSPO | Δ | SocialGPT | + GSPO | Δ |
| Vicuna-7B | 61.58 | 62.99 | +1.41 | 45.13 | 49.79 | +4.66 |
| Vicuna-13B | **66.70** | **69.23** | +2.53 | **65.12** | **66.19** | +1.07 |
| Llama2-7B | 31.91 | 34.07 | +2.16 | 36.71 | 38.04 | +1.33 |
| Llama2-13B | 37.86 | 41.27 | +3.41 | 42.74 | 48.39 | +5.65 |

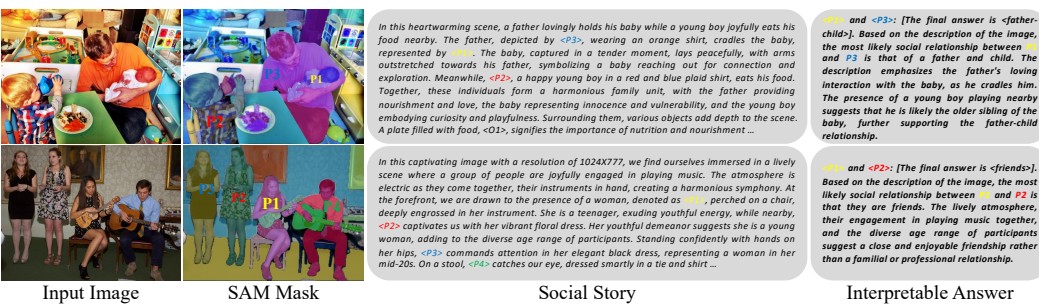

| Input Image | SAM Mask | Social Story | Interpretable Answer |

Figure 4: Visualization results of interpretability. We show the SocialGPT perception and reasoning process. We see that our model predicts correct social relationships with plausible explanations.

prompt and expectation segment contributes to the final performance by approximately 1.35% and 2.39%, respectively, and the context segment defining social relationship categories has a lesser contribution with a 0.4% accuracy difference. This is perhaps because the LLMs already have substantial knowledge of common social relationships.

## 5.3 Long Prompt Optimization with GSPO

As SocialGPT utilizes fixed prompt segments to instruct LLMs for social relation reasoning based on social stories, it might not be optimal with the static prompt design. Our GSPO further tunes the long prompt on the segment level for automatic performance improvements. Table 5 presents the results when applying GSPO on SocialGPT with various LLMs for reasoning, compared with the baseline zero-shot performance. Overall our segment-level prompt tuning with GSPO helps with the classification of all model variants. On PIPA the performance boost is about 2.38% on average, and on PISC it achieves a better gain with about 3.18% on average. These show the efficacy of the proposed GSPO algorithm to efficiently enhance prompt effectiveness. Out of the model variations, Vicuna-13B consistently outperforms other LLMs under our setup. The flexibility of SocialGPT in connecting with different reasoning models makes it more easily benefit from the latest advancements of LLMs without any heavy adaptation.

## 5.4 Qualitative Analysis

**Reasoning Process and Interpretability.** We illustrate the perception and reasoning process of SocialGPT as well as the final results in Figure 4. The people objects are fully segmented from VFMs and associated with symbols, which are then utilized to generate a coherent social story with clear references. By using LLMs for the reasoning on top of textual stories, SocialGPT not only outputs the correct social relations between different objects in the image but also provides plausible explanations behind the reasoning process.

**Generalization on Different Image Styles.** Previous supervised models on social relation recognition heavily rely on annotated images and relations in a specific domain. As a result, these models cannot generalize to unseen image types well. In contrast, our method does not have the limitation of being domain-specific. We apply SocialGPT to novel sketch and cartoon images with various social relations generated by GPT-4V, with results shown in Figure 5. As shown in the first example, the previous state-of-the-art model GR$^2$N [3] fails to generalize as it predicts the relation between $P_1$ and $P_2$ as colleagues, but SocialGPT correctly recognizes the classmate relation based on the social scene with detailed explanation.

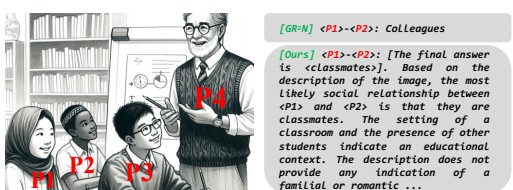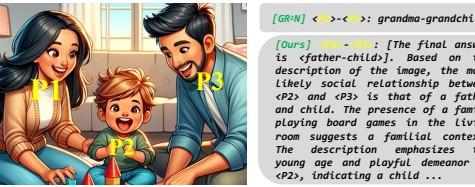

Figure 5: Results when applying SocialGPT to sketch and cartoon images. The images are generated by GPT-4V. Our method generalizes well on these novel image styles.

## 6  Conclusion

**Conclusion.** In this paper, we present SocialGPT, a modular framework with foundation models for social relation reasoning, which attains competitive zero-shot results while also providing interpretable explanations. Furthermore, we propose the GSPO for automatic prompt tuning, which further improves the performance. Our approach opens new avenues for exploring the synergy between vision and language models in high-level cognitive tasks and offers a promising direction for future advancements in the field of social relation recognition.

**Limitations and broader impacts.** Due to the modular nature of our approach, the performance of our method is constrained by the performance of the foundation models. If the segmentation model fails, or if the BLIP-2 model generates incorrect captions, or if the reasoning by LLMs is flawed, then our method is also prone to errors. Our method transforms visual problems into language-based reasoning, which could improve accessibility for visually impaired individuals. Meanwhile, our method also inherits biases from the foundation models, thus further research is needed to address them. Automatic classification of social relationships may lead to unintended negative consequences. To mitigate these risks, we can implement strategies such as fairness and bias checks, as well as promote transparent and responsible use of our technology.

## Acknowledgment

This research is supported in part by the NIH grant R01HD104969, NIH grant 1U01CA284207, and NSF award IIS-2239688.

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

## A More Implementation Details

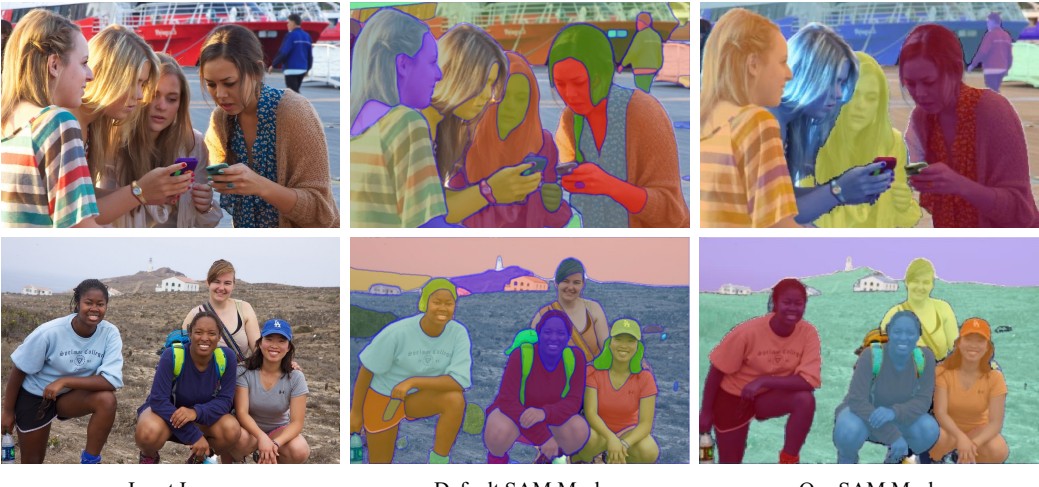

Input Image           Default SAM Mask           Our SAM Mask

Figure 6: The comparisons of the default SAM masks and our SAM masks.

In this paper, we employ SAM to automatically segment an image into multiple object masks, which we then use to generate dense captions. However, a challenge arises with SAM's default "segment everything" setting, as it tends to produce over-segmented and fine-grained masks. For instance, a person may be segmented into multiple fragments, including hair, face, hand, arm, and so on. Two examples illustrating this issue are presented in Figure 6. Creating meaningful captions for these subpart-level regions proves to be challenging and often leads to a loss of overall object perception. This is due to the fact that SAM generates three masks for each point prompt, corresponding to three semantic levels: whole, part, and sub-part. To address this issue, we adopted a two-stage SAM forward scheme. Initially, we employed SAM's default "segment anything" approach to obtain segmented masks, then retained the center points of each mask as point prompts for the second SAM forward pass. This ensures that as much as possible, objects in the image are not missed in the second SAM segmentation stage. For the second SAM segmentation stage, the points obtained from the first stage are used as point prompts, considering only the highest semantic level among SAM's three semantic levels. This approach minimizes over-segmentation and allows our method to focus on semantic at the object level. Subsequently, we apply NMS, threshold filtering, and post-processing to obtain high-quality object-level masks following SAM's methodology [17]. The resulting object masks for our method are displayed in Figure 6.

## B Prompts

**Social Story Generation.** We carefully designed the prompt to guide the LLMs in generating coherent and easily understandable social stories based on dense captions. The system prompt and user prompt are depicted in Figure 7. To ensure symbol-based referencing, we explicitly instruct LLMs not to rely on coordinates but instead to use symbols for reference. Additionally, we require the generated paragraphs to focus on social contexts.

**SocialPrompt on the PIPA dataset.** The PIPA dataset comprises 16 social relationship categories, including father-child, mother-child, grandpa-grandchild, grandma-grandchild, friends, siblings, classmates, loves/spouses, presenter-audience, teacher-student, trainer-trainee, leader-subordinate, band members, dance team members, sport team members, and colleagues. Figure 8 illustrates the prompt we utilized for the PIPA dataset in the zero-shot setting. We provided a detailed explanation for each category within the prompt. Furthermore, the SocialPrompt includes manually constructed in-context examples.

**SocialPrompt on the PISC dataset.** Figure 9 illustrates the SocialPrompt utilized in the PISC dataset, specifically in the zero-shot setting. The PISC dataset comprises 6 social relation categories:

[System Prompt]: You are an expert in generating only one naturally fluent and flawless paragraph based on a set of statements.

You must follow these rules:
- Illustrate the spatial relationship and depict the interaction between different people.
- Do not use any coordinate to describe.
- Must use symbols <O..> and <P..> when referring to objects and people.

[User Prompt]: Here we have 4-tuple [x1,y1,w,h] to depict the position of a box that frames the objects or persons, where [x1,y1] means the coordinate of the upper left corner of the box and [w,h] means the width and length of the box. The structure of people semantic is like "{<symbol>:[P..], <coordinate>:[x1,y1,w,h], <caption>:[caption text of the people], <age>:[age text of the people], <gender>:[gender text of the people]}". The structure of objects semantic is like "{<symbol>:[O..], <coordinate>:[x1,y1,w,h], <caption>:[caption text of the object]}". Generate only an informative and nature paragraph based on the given information (a,b,c,d) and following rules:

 a. Image Resolution: {width}X{height}
 b. Image Caption: {caption}
 c. Image Scene: {image caption scene}
 d. People and Objects Semantic: {region semantic}

There are some rules:
- Pay more attention to the people semantic, which have reference <P>.
- Depict the spatial relationships between individuals and objects, as well as the spatial relationships between people.
- Must use symbols <O..> and <P..> when referring to objects and people.
- Do not use coordinates [x1,y1,w,h], [x1,y1], [w,h] or numbers to show position information of each object.
- Pay more attention to the social scene and describe the social event in detail. Explain how each person and object contributes to the social event.
- No more than 15 sentences.
- Only use one paragraph.

Figure 7: The prompt used for social story generation. GPT-3.5 Turbo model is used for caption fusion. The system prompt lists some key rules and the user prompt details the task definition.

commercial, couple, family, friends, professional, and no-relation. We have also included the definitions of these six social relation categories within the prompt.

**SocialPrompt after GSPO.** Due to the time and effort-intensive nature of manually designing prompts, this paper introduces the Greedy Segment Prompt Optimization method. For each segment, we employ ChatGPT to generate multiple candidates. As for the in-context examples, we also randomly select several samples from the training dataset. Here, we employ Vicuna-7B [29] for training to obtain the optimized prompts. The optimized prompt on the PIPA dataset is illustrated in Figure 10, while that on the PISC dataset is shown in Figure 11.

[User Prompt]: As a social relations expert, you have the skill to accurately identify the category of social relationships portrayed in an image based on its text description. Your expertise covers 16 distinct types of social relationships, with each pair of individuals falling under one of these 16 categories. Using the provided information, you draw inferences to determine the most likely type of social relationship depicted in an image. Your final output should be one of 16 distinct types of social relationships, defined as follows: {<father-child>, <mother-child>, <grandpa-grandchild>, <grandma-grandchild>, <friends>, <siblings>, <classmates>, <lovers/spouses>, <presenter-audience>, <teacher-student>, <trainer-trainee>, <leader-subordinate>, <band members>, <dance team members>, <sport team members>, <colleagues>}.

Here is the definition:
1. <father-child>: The relationship between a father and his child, referring to a male who becomes the biological or legal father of one or more children.
2. <mother-child>: The relationship between a mother and her child, referring to a female who becomes the biological or legal mother of one or more children.
3. <grandpa-grandchild>: The relationship between a grandfather and his grandchild, referring to a male who becomes the grandfather of one or more grandchildren.
4. <grandma-grandchild>: The relationship between a grandmother and her grandchild, referring to a female who becomes the grandmother of one or more grandchildren.
5. <friends>: The relationship between two or more individuals who establish an intimate connection, usually based on shared interests, experiences, or backgrounds.
6. <siblings>: The relationship between two or more individuals who share the same parents or blood relations.
7. <classmates>: The relationship between students who study in the same class.
8. <lovers/spouses>: The romantic relationship between two individuals, which may include a marriage relationship.
9. <presenter-audience>: The relationship between a speaker and a group of listeners, where the speaker (usually a professional) delivers a speech or presentation to the audience, who may be viewers, listeners, spectators, or clients.
10. <teacher-student>: The relationship between a teacher and one or more students, where the teacher (usually a professional) imparts knowledge, skills, and values to the student.
11. <trainer-trainee>: The relationship between a trainer and one or more trainees, where the trainer imparts specific knowledge, skills, and techniques.
12. <leader-subordinate>: The relationship between a leader and their subordinates, where the leader holds a managerial position in an organization or institution, guiding and directing the activities of their subordinates.
13. <band members>: The relationship between musicians or singers who form a group to perform music together.
14. <dance team members>: The relationship between dancers who form a group to perform dance routines together.
15. <sport team members>: The relationship between athletes who form a team to compete in various sports.
16. <colleagues>: The relationship between individuals who work in the same organization or company.

****************************************************************
[1. image description]:
In this image with a resolution of 1024X768, a captivating scene unfolds on a sun-kissed beach. Captured in the frame are a woman and a young girl, their presence adding a sense of joy and tranquility to the serene surroundings. The young girl, denoted as P1, can be seen sitting on the sandy ground, her innocent curiosity shining through her bright eyes. Nearby, the woman, referred to as P2, gracefully bends her leg, taking in the beauty of the shoreline. The composition skillfully portrays the spatial relationship between individuals and objects, as well as the spatial relationships between people. Various objects enhance the scene's vibrant narrative. O1 depicts a suited man confidently holding a cell phone, exuding a professional aura. The focused attention on the people semantic continues with another shot of P3, the young girl leisurely seated on the powdery sand. Moving on to other objects that add charm and interest to the image, O2 presents a girl sitting with her legs crossed, while O3 showcases a content black cat resting on a wooden floor. A black and white dog proudly stands next to a white cat in O4, creating a delightful contrast. The visual narrative extends to O5, featuring a box adorned with a picture of two children, evoking a sense of innocence and playfulness. O6 offers a vibrant red and white bow set against a contrasting black background, injecting a touch of whimsy. The colorful ensemble continues with O7, capturing the striking image of a blue and black boat with a vibrant red bow. O8 showcases a white and pink table adorned with a picture of a woman, adding a feminine touch to the beach scene. Lastly, O9 and O10 portray a white house and a white garage respectively, both complemented by black roofs, completing the picturesque environment. Through careful attention to both the positions and the objects themselves, this image effectively conveys a sense of harmony and intrigue within the bounds of the beach setting.

[2. Question]:
What are the most likely social relationships between P1 and P2? Choose only one from {<father-child>, <mother-child>, <grandpa-grandchild>, <grandma-grandchild>, <friends>, <siblings>, <classmates>, <lovers/spouses>, <presenter-audience>, <teacher-student>, <trainer-trainee>, <leader-subordinate>, <band members>, <dance team members>, <sport team members>, <colleagues>}.

[3. Answer]: [The final answer is <mother-child>]. Based on the setting and the activities depicted, a likely inference could be the "mother-child" relationship. The image's narrative implies a protective and nurturing dynamic between the woman and the young girl, which is characteristic of a mother-child relationship. Considering that P1 is a young girl and P2 is a woman, and they are together at the beach, one plausible inference could be the "mother-child" relationship. The beach setting could suggest a family outing, and the age difference suggests a parent-child dynamic.
****************************************************************

Figure 8: The SocialPrompt on the PIPA dataset.

[User Prompt]: Possessing expertise in social relations, you hold the proficiency to correctly categorize the social relationships depicted in an image, by analyzing its textual description. Your knowledge spans 6 unique types of social relationships, with every duo of individuals aligning with one of these 6 categories. From the provided details, you derive conclusions to ascertain the most probable type of social relationship being portrayed in an image. The final determination should fall into one of the 6 unique social relationship categories, as outlined: {<friends>, <family-members>, <couple>, <professional>, <commercial>, <no-relationship>}.

Here is the definition:
1. <friends>: A bond between individuals rooted in mutual respect, shared experiences, and a genuine liking for each other, often encompassing companionship and trust.
2. <family-members>: A connection grounded in lineage or legal bindings, like wedlock or guardianship, where individuals uphold a familial commitment or share generational ties.
3. <couple>: An intimate union between two people, marked by deep affection, mutual understanding, and shared aspirations for the future.
4. <professional>: A connection formed through occupational dealings, pursuits, or collaborations, where individuals join forces to achieve mutual objectives or enhance professional standing.
5. <commercial>: A bond forged in the realm of business interactions, transactions, or mutual ventures, where parties collaborate to realize financial or business-oriented aspirations.
6. <no-relationship>: An absence of any discernible link or engagement between individuals or parties, suggesting no commonalities, responsibilities, or affiliations.

****************************************************************
[1. image description]:
In the bustling scene of a parade, a group of police officers on horseback captivates the attention of the crowd. Among them, a woman wearing a hat and scarf (<P1>) stands tall, exuding confidence. Close by, a woman in a purple scarf and black jacket (<P2>) commands authority as a police officer. A police officer in a hat and sunglasses (<P3>) adds an air of mystery to the scene. In a surprising twist, a man in a police uniform rides a skateboard (<P4>), showcasing his youthful spirit. Another man in a police uniform, wearing sunglasses (<P5>), exudes a sense of coolness. A woman in a white hat and scarf (<P6>) beams with joy, adding a touch of warmth to the parade. The presence of horses with saddles and bridles (<O1>, <O2>, <O3>) symbolizes the traditional and noble nature of the police force. A group of people standing in a line (<O4>) signifies the unity and camaraderie among the officers. A blue and yellow police vest with the words "Washington Police Department" (<O5>) proudly represents the force. A man in a jacket and jeans standing against a black background (<O6>) adds an element of intrigue. A person holding up a bunch of stickers (<O7>) suggests the engagement of the crowd. A man in a black jacket and white scarf (<O8>) adds a touch of style to the event. A blue and white striped chair with a matching back (<O9>) provides a resting place for weary officers. Lastly, the flag of France displayed on a flagpole (<O10>) symbolizes the international cooperation and solidarity within the police force. Together, these individuals and objects create a vibrant and dynamic atmosphere, showcasing the dedication and diversity of the police officers in this parade.

[2. Question]:
What are the most likely social relationships between P5 and P6? Choose only one from {<friends>, <family-members>, <couple>, <professional>, <commercial>, <no-relationship>}.

[3. Answer]: [The final answer is <professional>]. The description portrays a scene of a parade where a group of police officers, including P5 and P6, are participating. They are both described as police officers, indicating a professional relationship. The focus of the description is on their roles and presence in the parade, suggesting a shared professional connection rather than a personal or romantic one. There is no evidence to suggest a familial, commercial, or friendship relationship between P5 and P6. Therefore, the most likely social relationship between them is a professional one.
****************************************************************

Figure 9: The SocialPrompt on the PISC dataset.

[User Prompt]:In your role as a specialist in social relations, you possess the capability to precisely determine the nature of social relationships shown in an image from its textual description. The range of your expertise encompasses 16 unique categories of social relationships, with each duo of individuals categorized under one of these. From the information given, you make deductions about the probable type of social relationship an image displays. The relationship type you conclude should be among the following 16 unique categories: {<father-child>, <mother-child>, <grandpa-grandchild>, <grandma-grandchild>, <friends>, <siblings>, <classmates>, <lovers/spouses>, <presenter-audience>, <teacher-student>, <trainer-trainee>, <leader-subordinate>, <band members>, <dance team members>, <sport team members>, <colleagues>}.

Here is the definition:
1. <father-child>: The bond between a father and his child, characterized by a male being the biological or legal guardian to one or more children.
2. <mother-child>: The bond between a mother and her child, embodied by a female being the biological or legal guardian to one or more children.
3. <grandpa-grandchild>: The bond between a grandfather and his grandchild, depicted by a male being the grandfather to one or more grandchildren.
4. <grandma-grandchild>: The bond between a grandmother and her grandchild, depicted by a female being the grandmother to one or more grandchildren.
5. <friends>: The bond between two or more individuals who foster a close connection, often stemming from common interests, shared experiences, or similar backgrounds.
6. <siblings>: The bond between two or more individuals who have common familial ties through either biological or legal parentage.
7. <classmates>: The bond between students who share the academic journey in the same class setting.
8. <lovers/spouses>: The romantic bond between two individuals, encompassing a union that may extend to a marital relationship.
9. <presenter-audience>: The interactive bond between a speaker and a group of listeners, wherein the speaker, often a professional, delivers content or messages to the attentive audience.
10. <teacher-student>: The educational bond between a teacher and one or more students, where the teacher, often a professional, disseminates knowledge, skills, and values to the student.
11. <trainer-trainee>: The instructional bond between a trainer and one or more trainees, with the trainer providing specific knowledge, skills, and techniques.
12. <leader-subordinate>: The hierarchical bond between a leader and their subordinates, where the leader, in a managerial position, navigates and orchestrates the activities of the subordinates within an organization or institution.
13. <band members>: The creative bond between musicians or singers who unite to create and perform music as a collective.
14. <dance team members>: The rhythmic bond between dancers who collaborate to choreograph and perform dance routines as a unit.
15. <sport team members>: The competitive bond between athletes who amalgamate into a team striving to achieve success in various sporting events.
16. <colleagues>: The professional bond between individuals who share a common working environment within an organization or company.

****************************************************************
[1. image description]:
In a meeting room, two men in suits are engaged in a professional discussion. The first man, <P1>, stands on the left side of the room, wearing a suit with a red tie. He is in his mid-50s and has a warm smile on his face. The second man, <P2>, stands on the right side, wearing a suit and tie. He is in his early 20s and is gesturing with his hands as he speaks. In front of them, there is a red chair, <O2>, which adds a touch of color to the scene. Behind the men, there is a wall with the words "the best way to get your," <O1>, emphasizing the importance of the meeting. On the wall, there is also a red heart, <O3>, symbolizing passion and dedication. Additionally, there is a red beret with black trim, <O4>, placed on a nearby table, adding a touch of style to the room. On the floor, there are a pair of red shoes, <O5>, which complement the overall aesthetic. The combination of these objects creates a visually appealing and professional atmosphere for the meeting. The interaction between the two men, <P1> and <P2>, is evident as they shake hands, symbolizing mutual respect and collaboration. Overall, this scene captures a significant moment of interaction and exchange between two individuals in a professional setting, with the objects enhancing the ambiance and adding depth to the overall composition.

[2. Question]:
What are the most likely social relationships between P1 and P2? Choose only one from {<father-child>, <mother-child>, <grandpa-grandchild>, <grandma-grandchild>, <friends>, <siblings>, <classmates>, <lovers/spouses>, <presenter-audience>, <teacher-student>, <trainer-trainee>, <leader-subordinate>, <band members>, <dance team members>, <sport team members>, <colleagues>}.

[3. Answer]: [The final answer is <colleagues>]. Based on the description of the image, the most likely social relationship between P1 and P2 is that they are colleagues. The setting of a meeting room and their professional attire suggest a work-related context. The description emphasizes their professional discussion and the interaction between them, indicating a professional relationship rather than a personal or familial one.
****************************************************************

Figure 10: The prompt after GSPO on the PIPA dataset.

[User Prompt]: Possessing expertise in social relations, you hold the proficiency to correctly categorize the social relationships depicted in an image, by analyzing its textual description. Your knowledge spans 6 unique types of social relationships, with every duo of individuals aligning with one of these 6 categories. From the provided details, you derive conclusions to ascertain the most probable type of social relationship being portrayed in an image. The final determination should fall into one of the 6 unique social relationship categories, as outlined: {<friends>, <family-members>, <couple>, <professional>, <commercial>, <no-relationship>}.

Here is the definition:
1. <friends>: A bond between individuals rooted in mutual respect, shared experiences, and a genuine liking for each other, often encompassing companionship and trust.
2. <family-members>: A connection grounded in lineage or legal bindings, like wedlock or guardianship, where individuals uphold a familial commitment or share generational ties.
3. <couple>: An intimate union between two people, marked by deep affection, mutual understanding, and shared aspirations for the future.
4. <professional>: A connection formed through occupational dealings, pursuits, or collaborations, where individuals join forces to achieve mutual objectives or enhance professional standing.
5. <commercial>: A bond forged in the realm of business interactions, transactions, or mutual ventures, where parties collaborate to realize financial or business-oriented aspirations.
6. <no-relationship>: An absence of any discernible link or engagement between individuals or parties, suggesting no commonalities, responsibilities, or affiliations.

**************************************************************
[1. image description]:
In this scene at an airport, a woman with a black bag stands in the center, surrounded by a diverse group of people and objects. To her left, a person in a green shirt and jeans, identified as <P1>, leans against a metal box, while a girl in a purple shirt and jeans, identified as <P2>, stands nearby. On the right, a man in a black shirt and red hat, identified as <P3>, carries a black suitcase with yellow tape, while a man in a green and white striped shirt, identified as <P4>, holds a piece of luggage with a tag attached. Behind them, a young adult man, identified as <P5>, stands in front of a black background, wearing a tan jacket. In the foreground, a woman in a blue and white outfit, identified as <P6>, jumps energetically. The objects in the scene include a woman bending over to pick up a frisbee, a man standing on a metal box, a suitcase with a green and yellow stripe, a black and blue bird with a long tail, a tan leather briefcase with a strap, a black wallet with a red and yellow stripe, a man in a black shirt and red tie, and a man standing in front of a window. Each person and object contributes to the bustling atmosphere of the airport, with the individuals engaged in various activities and the objects representing the luggage and personal belongings of the travelers.

[2. Question]:
What are the most likely social relationships between P4 and P5? Choose only one from {<friends>, <family-members>, <couple>, <professional>, <commercial>, <no-relationship>}.

[3. Answer]: [The final answer is <no-relationship>]. The description portrays a scene at an airport where a diverse group of individuals is present. There is no specific indication of a social relationship between P4 and P5. They are simply described as individuals standing near each other in the scene, without any explicit interaction or connection. Therefore, the most likely social relationship between P4 and P5 is no relationship.

**************************************************************

Figure 11: The prompt after GSPO on the PISC dataset.

