# OpenReview forum: "SocialGPT: Prompting LLMs for Social Relation Reasoning via Greedy Segment Optimization"
_NeurIPS.cc/2024/Conference — NeurIPS 2024 poster_

### Official Review · Reviewer_aaEq · 2024-07-10

**Soundness:** 4
**Presentation:** 4
**Contribution:** 3
**Rating:** 7
**Confidence:** 4

**Summary:**

Paper proposes a pipeline method of orchestration pre-trained foundation models to solve the social relationship classification problem. It uses vision models to extract information in text about the scene in the form of caption. Relevant information, i.e. age, gender, general description, of individual persons and objects are also extracted in text form via instance segmentation + masking + captioning. The generated text are then further converted to Social Story with a LLM. With the novel prompt engineering method, GSPO, another LLM will then generate the social relationship from the Social Story.

Experimental results on the challenging benchmarks, PIPA and PISC, indicates its strong performance with zero-shot setup. Extensive ablation studies were also done to evaluate the contributions of the various components. In particular, it clearly showed the merits of the "Social Story" design.

**Strengths:**

Paper proposed a novel method to solve the challenging social relationship classification problem. The proposed method cleverly combine several state-of-the-art foundation models in a logical, intuitive, and yet non-obvious design to achieve state-of-the-arts experimental results.

**Weaknesses:**

1. Besides the clever design of the pipeline, the direct technical contributions is slightly on the weaker side as there is no obvious technical breakthrough. The proposed GSPO appears to be the main new technique introduced. However, I am not an expert in this area and will defer to other reviewers on its technical novelty and merits.

2. (minor) The use of the generic semantic segmentation model (SAM) may not be the optimal choice. There are much stronger Human Instance Segmentation methods which can replace the paper's custom SAM method. Such methods are specifically trained on person dataset to handle various challenging scenarios unique to human segmentation, e.g. heavy occulsion, human-like objects (e.g. maniquinn).

Ling, E., Huang, D., & Hur, M. (2022). Humans need not label more humans: Occlusion copy & paste for occluded human instance segmentation. BMVC.

**Questions:**

1. Will/has the authors consider using pairwise attributes, besides the individual person attributes. E.g. relative age between pairs (older/younger), same/different clothings for the model?

2. Why are only 2 attributes (age/gender) used for the person instance? In prior works, other attributes such as wearing uniform are important attributes for certain type of social relationship, e.g. team members?

**Limitations:**

(minor) There may be some unintended negative consequences of automatic classification of social relationship.

---

> ### Author Rebuttal · Authors · 2024-08-07
>
> Thank you for the constructive feedback and the positive assessment of our work! Below, we detail our responses to the review concerns.
>
> **W1: Technical novelty of GSPO**
>
> Thank you for acknowledging the "clever" design of our pipeline. We agree that our major technical contribution lies in our proposed GSPO. Our SocialGPT transforms a visual classification task into a generative task of LLMs, presenting unique challenges in long prompt optimization that cannot be effectively tackled by existing methods. To address this issue, our GSPO method innovatively performs a greedy search utilizing gradient information at the segment level. The experimental results further validate the effectiveness of our proposed GSPO algorithm.
>
> **W2: Human instance segmentation methods**
>
> Thank you for your suggestion! In response, we incorporated the model suggested in [1] into our pipeline to directly compare it with our custom SAM method. The experimental results, presented in Table R4-1, indicate that the SAM model outperforms [1], achieving higher accuracy. The reason for SAM's superior performance is its ability to segment not only human figures but also non-person objects. This capability is particularly beneficial for social relation recognition, which relies on contextual object cues in addition to human figures. We will cite [1] and detail these additional experiments in the revised manuscript to provide a comprehensive view of our methodological choices.
>
> Table R4-1. Results with different segmentation methods
> |Segmentation| [1] |SAM|
> |-|-|-|
> |Accuracy (%)|57.26|64.10|
>
> [1] Humans need not label more humans: Occlusion copy & paste for occluded human instance segmentation, BMVC 2022.
>
> **Q1: Pairwise attributes**
>
> Thank you for your valuable suggestion! We conducted experiments by adding one more relative attribute to our method. Specifically, we extended the queries to BLIP-2 to include not only individual age and gender attributes but also relative age or clothing differences between pairs.
>
> The experimental results are summarized in Table R4-2. While the inclusion of relative age did not significantly alter the performance, suggesting that our original age attribute effectively captures relative age information, the addition of clothing attributes enhanced our accuracy by 1.03%. This indicates that incorporating complementary attributes can indeed enhance the performance of our model.
>
> Table R4-2. Results with more attributes
> |Attributes| Ours | + relative age | + clothing |
> |-|-|-|-|
> |Accuracy (%)|64.10|63.98|65.13|
>
> **Q2: Other attributes**
>
> Thank you for your question! As demonstrated in Table R4-2, incorporating additional attributes such as clothing information indeed enhances the performance of our framework. We would like to highlight that the primary focus of our paper is on developing a foundational model-based framework for zero-shot social relation reasoning and addressing the long prompt optimization issue within this framework. The decision on the number and type of attributes to generate task-oriented captions was driven by the desire to establish a simple baseline, which is not the main focus of our work. We initially selected age and gender as they are commonly referenced attributes that provide foundational insights for our zero-shot social relation reasoning baseline.
>
> While we acknowledge the potential benefits of incorporating more diverse attributes, we opted to limit the scope of this initial study. We believe that exploring a broader array of attributes presents an exciting direction for future research, potentially leading to further improvements in performance.
>
>
> **L1: Unintend negative consequences**
>
>
> Thank you for raising this important concern. We recognize that automatic classification of social relationships can indeed lead to unintended negative consequences. To address this, we will expand our discussion in the broader impacts section of the paper.
>
> In this expanded discussion, we plan to outline potential risks and propose mitigation strategies, such as implementing fairness and bias checks, and promoting transparent and responsible usage of our technology. We appreciate this suggestion and believe that addressing these concerns thoroughly will enhance the quality and ethical standing of our work.

---

> > ### Author Response · Authors · 2024-08-13
> > **Looking forward to your feedback**
> >
> > Dear reviewer,
> >
> > Thank you for the comments on our paper. We have responded to your initial comments. We are looking forward to your feedback and will be happy to answer any further questions you may have.

---

### Official Review · Reviewer_RdU3 · 2024-07-12

**Soundness:** 3
**Presentation:** 3
**Contribution:** 3
**Rating:** 5
**Confidence:** 3

**Summary:**

This paper introduces SocialGPT, a modular framework for social relation reasoning that integrates the perception capabilities of Vision Foundation Models (VFMs) with the reasoning capabilities of Large Language Models (LLMs). To optimize prompts, the authors propose GSPO, a segment-based optimization algorithm for automated prompt tuning. Extensive empirical results validate the effectiveness of SocialGPT both quantitatively and qualitatively. GSPO consistently enhances SocialGPT's performance across various LLMs, and case studies demonstrate the framework's generalizability and interpretability.

**Strengths:**

- The paper is well-organized, with a logical flow and clear explanations of each step.
- The proposed SocialGPT framework innovatively combines perception from VFMs with reasoning from LLMs, achieving competitive zero-shot performance and offering potential explanations for its reasoning process.
- Extensive experiments, ablation studies, and case studies comprehensively evaluate the framework's effectiveness.

**Weaknesses:**

Section 3.2 mentions that using precise coordinates can pose challenges for LLM numerical reasoning. However, it appears in Figure 3 that the objects' positional relations in the social story are inferred from numeric coordinates provided in the dense captions with symbols. Does this coordinate-based inference lead to similar numerical reasoning challenges? Additionally, how are relative positional relations conveyed here using referral symbols?

**Questions:**

Please see the weaknesses section above.

**Limitations:**

The authors have adequately addressed the limitations.

---

> ### Author Rebuttal · Authors · 2024-08-07
>
> Thank you for the constructive feedback and the positive assessment of our work! We are happy that you find our framework **innovative** and our experimental evaluation **comprehensive**. Below, we detail our responses to the review concerns.
>
> **W1-1**
>
> > Does this coordinate-based inference lead to similar numerical reasoning challenges?
>
> Thank you for the question! Indeed, using precise coordinates for social reasoning in LLMs poses substantial challenges as it requires the model to understand spatial relationships and perform social reasoning simultaneously, which can be complex.
>
> To mitigate these challenges, we adopt a two-step approach in our methodology: first, we generate textual social stories by converting coordinates into descriptive textual spatial information; subsequently, we use these stories for social reasoning. This decomposition simplifies the cognitive load on the LLM by separating spatial understanding from social reasoning.
>
> Recent research [1][2] on LLMs supports this approach: they have demonstrated that breaking down complex tasks into simpler, manageable sub-tasks for multi-step reasoning significantly enhances LLM performance by reducing cognitive demands.
>
> Our empirical results further validate this method. As shown in Table 2 of our paper, the performance of our proposed method surpasses that of the coordinate-based method, underscoring the effectiveness of our strategy in reducing the difficulties associated with numerical reasoning in social contexts.
>
> [1] Decomposed prompting: A modular approach for solving complex tasks, ICLR 2023
>
> [2] Least-to-most prompting enables complex reasoning in large language models, ICLR 2023
>
> **W1-2**
> > How are relative positional relations conveyed here using referral symbols?
>
>
> Thank you for the question! To address this issue, we instruct the LLM to describe spatial relationships among objects and people using textual descriptors instead of relying on precise coordinates. This is achieved by using specific prompts that focus on textual descriptions rather than numerical data.
>
> For instance, we use prompts such as: "Depict the spatial relationships between individuals and objects, as well as the spatial relationships between people", "Must use symbols <O..> and <P..> when referring to objects and people", and "Do not use coordinates [x1,y1,w,h], [x1,y1], [w,h] or numbers to show position information of each object". These prompts guide the LLM to generate social stories that describe spatial relationships. The full set of prompts used for generating these social stories is detailed in Figure 7 of our paper, providing a comprehensive view of our methodology.
>
> Figure 3 in our main paper shows an example of how the generated social stories describe the relative positional relations: "At the center of the image, a young <P1> boy", "To his right, a middle-aged <P2> man", "On the left side of the frame, a <P3> woman". We observe that LLM can reason the relative positional relations with our carefully designed prompts.

---

> > ### Author Response · Authors · 2024-08-13
> > **Looking forward to your feedback**
> >
> > Dear reviewer,
> >
> > Thank you for the comments on our paper. We have responded to your initial comments. We are looking forward to your feedback and will be happy to answer any further questions you may have.

---

> > > ### Comment · Reviewer_RdU3 · 2024-08-13
> > >
> > > Thanks for your response. My questions have been well addressed, and I will maintain my score.

---

> > > > ### Author Response · Authors · 2024-08-13
> > > > **Thank you**
> > > >
> > > > We are glad that all your questions have been well addressed. Thank you again for your positive assessment and for taking the time to review our work. Your invaluable feedback has been pivotal in improving our paper.

---

### Official Review · Reviewer_zGW2 · 2024-07-13

**Soundness:** 3
**Presentation:** 3
**Contribution:** 2
**Rating:** 4
**Confidence:** 4

**Summary:**

This paper proposes a framework called SocialGPT for social relation reasoning, which combines vision foundation models and large language models. A greeedy segment prompt optimization methods is also proposed to prompt LLM. Experimental results show the effectiveness of the proposed method.

**Strengths:**

---The paper is well organized and written.

---The idea of combining VFMs and LLMs is reasonable.

**Weaknesses:**

--- The paradigm of using VLMs for perceiving and LLMs for reasoning is currently a common solution for multimodal tasks. The main difference of this paper seems to be the use of a generated social story as the representation of visual content. As stated by the authors, LLMs perform best when working with human-readable natural language and often struggle with arithmetic reasoning tasks, which is why they design an additional process to generate social stories. However, the generation of social stories is also done by LLMs, which also suffer from the above difficulties.

--- The authors propose a candidate set consisting of alternative prompts for each segment and select the best-performing prompt from their combination. The final prompt is obtained by selection rather than generation, which limits the upper bound of the performance on the manually collected candidate set.

--- The function of SAM is to distinguish individuals in the image and obtain their coordinates. However, in the social story generation phase, the LLM (Large Language Model) discards the coordinates, retaining only the semantic information and losing the positional information. Conducting social relationship reasoning purely based on semantics may be insufficient. For example, in Figure 2, the social relationship is identified as a sibling relationship (brother and sister), but there are two boys in the image, both fitting the given description of "stands out in his vibrant red and green striped pajamas," making it unclear which individual P1 refers to.

**Questions:**

--- Is the design of using LLMs for social story generation optimal, and why? Also, have the authors tried other approaches to generate social stories from dense captions instead of using LLMs?

--- In the part of reasoning with large language models, the social relation reasoning prompt is artificially divided into four partitions: System, Expectation, Context, and Guidance, but the motivation and reasonableness of such a design is not elaborated in the paper.

**Limitations:**

Please see the weakness and limitations.

---

> ### Author Rebuttal · Authors · 2024-08-07
>
> Thanks for the comments! Below we address the detailed questions. We hope that our responses will reflect positively on your final decision.
>
> **W1-1: Common solution**
>
> We are fully aware that leveraging foundation models for vision tasks is a growing trend, which also motivates our work. We would like to emphasize that while our SocialGPT provides a simple zero-shot social relation baseline using foundation models, our major technical contribution lies in our GSPO. It addresses the long prompt optimization issue, automatically optimizing the SocialPrompt.
>
> **W1-2: Difficulties for social story generation**
>
> Recent studies [1][2] on LLMs have shown that decomposing complex tasks, such as mathematical reasoning, into sub-tasks for multi-step reasoning, effectively reduces difficulties, and improves performance. Therefore, instead of directly using LLMs for social reasoning with raw, dense captions, we first integrate these captions into a coherent social story. This story conveys positional information through text rather than numerical coordinates, and then we perform social reasoning with the social stories. This two-step decomposition has reduced the reasoning difficulties and achieved higher performance. We conducted ablation experiments on social stories, and the results in Table R2-1 show that the introduction of social stories greatly improved performance.
>
> Table R2-1. Ablation on social story
> |social story|PIPA|PISC|
> |-|-|-|
> |without|45.31|37.42|
> |with|61.58|45.13|
>
> [1] Decomposed prompting: A modular approach for solving complex tasks, ICLR 2023
>
> [2] Least-to-most prompting enables complex reasoning in large language models, ICLR 2023
>
> **W2: Upper bound**
>
> There are many works on automatic prompt tuning for LLMs; one popular direction is performing greedy exhaustive searches or selections over tokens. These methods [3][4] are well-recognized in the NLP field. They typically perform well and do not face any upper-bound issues. For example, AutoPrompt [3] identifies a candidate set of the top-k tokens and performs greedy selection, which has garnered over 1500 citations. Our GSPO further proposes performing a greedy search by utilizing gradient information at the segment level to address the long prompt optimization issue.
>
> We want to clarify that, besides the example segment, the candidate sets for other segments are generated by ChatGPT and are not manually collected. For the example segment, the candidates can be chosen from the entire training set, thus GSPO is also not constrained by a limited candidate set. The results in Table 5 of our paper also validate the effectiveness of our GSPO.
>
> [3] Autoprompt: Eliciting knowledge from language models with automatically generated prompts, EMNLP 2020
>
> [4] Automatically Auditing Large Language Models via Discrete Optimization, ICML 2023
>
> **W3: Losing positional information**
>
> Sorry for the misunderstanding. We want to clarify that in our social story generation phase, we explicitly instruct LLMs to preserve positional information, rather than lose it. We discard only the numerical coordinates and use textual sentences to describe positional information. The corresponding prompts
> > Illustrate the spatial relationship and depict the interaction between different people
>
> and
> > Depict the spatial relationships between individuals and objects, as well as the spatial relationships between people
>
> are shown in Figure 7 of our paper.
>
> Additionally, our generated social story could include very detailed positional cues, as exemplified in Figure 3 of our paper: "At the center of the image, a young <P1> boy", "To his right, a middle-aged <P2> man", "On the left side of the frame, a <P3> woman". As for the example in Figure 2, it actually contains weaker positional information due to the performance fluctuations of LLMs: "gathered", "Sitting beside him is a girl, <P2>". While this is not sufficient to reconstruct the original image details, it is adequate for LLM reasoning, which is our primary requirement.
>
> To further verify the effectiveness of our position-related prompts in the social story generation phase, we compared scenarios with and without these prompts. The results in Table R2-2 show the effectiveness of our positional prompts.
>
> Table R2-2. Ablation on positional prompts
> |Method|w/o pos|w. pos|
> |-|-|-|
> |Accuracy (%)|58.32|61.58|
>
> **Q1: Other approaches to generate social stories**
>
> We considered and evaluated two alternative approaches alongside our LLM-based method: 1) BLIP-2: instructing BLIP-2 to generate social stories. 2) Concatenation: combining all dense captions into a single paragraph. Table R2-3 shows that the LLM-based method attains the best performance.
>
> Table R2-3. Results of different social story generation approaches.
> |Method|BLIP-2|Concatenation|LLM|
> |-|-|-|-|
> |PIPA|43.85|45.31|61.58|
> |PISC|40.17|37.42|45.13|
>
> **Q2: Four partitions**
>
> The Four partitions are well motivated and supported by the recent research progress in LLM. Research [5] shows the system prompt "establishes the foundational context for the model’s responses". Study [6] demonstrates that assigning roles to LLMs can enhance their performance, which illustrates the importance of the Expectation prompt. According to [7], incorporating background knowledge benefits LLMs reasoning, strongly motivating the use of the Context prompt. As for the Guidance prompt, using in-context examples has become a common practice to improve LLM performance [8].
>
> Besides the above evidence, Table 2 of our paper provides a detailed ablation, validating their effectiveness.
>
> [5] Jailbreaking GPT-4v via self-adversarial attacks with system prompts, arXiv:2311.09127 (2023).
>
> [6] Better Zero-Shot Reasoning with Role-Play Prompting, NAACL 2024.
>
> [7] Knowledge-Augmented Language Model Prompting for Zero-Shot Knowledge Graph Question Answering, ACL Workshop 2023.
>
> [8] Chain-of-Thought Prompting Elicits Reasoning in Large Language Models, NeurIPS 2022.

---

> > ### Author Response · Authors · 2024-08-12
> > **Looking forward to your feedback**
> >
> > Dear reviewer,
> >
> > Thank you for the comments on our paper. We have responded to your initial comments. We are looking forward to your feedback and will be happy to answer any further questions you may have.

---

> > > ### Author Response · Authors · 2024-08-14
> > > **Looking forward to your feedback before the end of discussion period**
> > >
> > > Dear reviewer,
> > >
> > > Thank you very much for your valuable feedback. As the discussion period is coming to an end, we kindly wish to inquire if our response has fully addressed your concerns. Please let us know if there are any additional concerns or questions.

---

### Official Review · Reviewer_9pM3 · 2024-07-13

**Soundness:** 2
**Presentation:** 3
**Contribution:** 2
**Rating:** 4
**Confidence:** 3

**Summary:**

This manuscript introduces SocialGPT, a modular framework designed to enhance social relation reasoning by combining Vision Foundation Models (VFMs) and Large Language Models (LLMs). SocialGPT utilizes VFMs to convert image content into a textual social story, followed by LLMs performing text-based reasoning. The paper further introduces the Greedy Segment Prompt Optimization (GSPO) algorithm to optimize prompts for LLMs, addressing the challenges of long prompt optimization. The proposed method achieves competitive zero-shot results on social relation recognition tasks and offers interpretable answers.

**Strengths:**

- The GSPO algorithm provides an efficient method for optimizing long prompts, significantly improving the performance of LLMs in social relation reasoning tasks.
- SocialGPT achieves competitive zero-shot results on PIPA and PISC datasets, demonstrating the effectiveness of the proposed approach without additional model training.
- By leveraging LLMs for reasoning, SocialGPT can generate language-based explanations for its decisions, enhancing the interpretability of the results.

**Weaknesses:**

- The approach involves substantial computational resources for both the perception and reasoning phases, potentially limiting accessibility and scalability for some users.
- The experiments, while promising, are primarily conducted on two datasets. Further testing on a broader range of datasets and tasks would strengthen the generalizability of the findings.
- The method assumes that the visual context provided by VFMs is sufficiently detailed and accurate, which might not always hold true in diverse real-world scenarios.
- The compatibility of the proposed method seems to be limited; Table 5 implies that LLaMA2-based SocialGPT performs very poorly compared to Vicuna. The proposed framework may work only for specific types of models.

**Questions:**

- How can we evaluate generated social stories? It would be great if the authors could show how GSPO improves the quality of generated social stories.
- How GSPO can be performed without the ground-truth answer? The current formulation in section 4 seems to require the ground-truth to define the loss objective.
- What are the differences between the social story of SocialGPT and social relationships used in baselines? I feel that Image-based text explanation is not new.

**Limitations:**

See the weakness section and question above.

---

> ### Author Rebuttal · Authors · 2024-08-07
>
> Thank you for your valuable comments! Please find our responses to specific queries below. We hope that our responses will reflect positively on your final decision.
>
> **W1: Substantial computational resources**
>
> Our method leverages multiple foundation models, which may initially appear computationally intensive compared to traditional social relation methods that use dedicated networks. However, the paradigm shift introduced by foundation models offers a significant advantage: they enable solving multiple downstream tasks with just one set of models using different prompts, rather than training a separate model for each task. Using a single set of foundation models for diverse tasks, as demonstrated by the success of ChatGPT, is not only economical but also enhances scalability and accessibility. Our method facilitates deployment in scenarios where intelligent agents are expected to handle multiple tasks simultaneously. Therefore, our approach ultimately presents a more scalable and accessible solution, aligning with the future directions of intelligent systems deployment.
>
> **W2: The experiments are conducted on two datasets**
>
> Our choice to conduct experiments on two datasets (PIPA and PISC) aligns with common practices in the field. Previous SOTA methods such as Dual-Glance, GRM, SRG-GN, GR2N, and TRGAT have utilized these datasets, either individually or in combination, as benchmarks.
>
> As there are no other commonly used social relation datasets, following the suggestion, we further collected and annotated a small dataset consisting of 100 samples from diverse real-world scenarios on our own. We then directly tested the zero-shot performance on this dataset, and the results below show that our method significantly outperforms the previous open-sourced SOTA method, GR2N.
>
> Table R1-1. Results on a new dataset
> |Method |GR2N|Ours(GPT-3.5)|Ours(Vicuna-13B)|
> |-|-|-|-|
> |Accuracy (%)|48.0|63.0|62.0|
>
> **W3: Detailed and accurate visual context**
>
> While converting all information from an image into text with VFMs can be challenging, obtaining **social-related** visual information sufficient for later reasoning is easier and feasible. Our SocialGPT introduces several designs to facilitate such a process: generating dense captions with SAM to cover all objects, inquiring social-related questions through BLIP-2 to obtain task-oriented captions, and generating social stories to depict the image from a social relations perspective. With these designs, our SocialGPT, without any fine-tuning on social relation datasets, achieves better performance on the PIPA and PISC datasets compared with SOTA methods that are specifically trained on these social datasets. This illustrates the effectiveness of the proposed framework. Furthermore, compared with previous methods trained on social datasets, foundation models usually provide better generalization in diverse real-world scenarios, which is also validated by the results in Table R1-1.
>
> **W4: The compatibility**
>
> The performance of our method is directly influenced by the reasoning ability of LLMs, as we use LLMs for final social relation prediction. Table R1-2 shows that with a more advanced LLM version, Llama 3 achieves performance comparable to that of other LLMs. These findings illustrate that our framework is adaptable and can effectively leverage advancements in LLM technology, showing the compatibility of our method.
>
> Table R1-2. Results with different LLMs
> |LLM|GPT-3.5|Vicuna-13B|Llama-3-8B|
> |-|-|-|-|
> |PIPA|64.1|66.70|67.82|
> |PISC|53.43|65.12|65.34|
>
> **Q1: Evaluate social stories**
>
> To evaluate the quality of generated social stories, we adopt another LLM (GPT-4) as the judge. We input two different social stories for the same image into this judge LLM and ask it to decide which one describes a more vivid and detailed social story. Our social stories are generated using the prompts shown in Figure 7 of our main paper. We compare our results with a baseline method that instructs LLMs to fuse dense captions without our carefully designed prompts. The results in Table R1-3 show that our generated social stories are of high quality.
>
> Table R1-3. Social story evalutaion
> |Method|Baseline|Ours|
> |-|-|-|
> |Preference|0.13|0.87|
>
> > It would be great if the authors could show how GSPO improves the quality of generated social stories.
>
> We want to clarify that GSPO is not used for improving social stories; instead, it is used to find a better SocialPrompt (see Figure 2). This improved prompt can better instruct LLMs in social relation reasoning. We have included the SocialPrompts before and after using GSPO in Figures 9-11 of our main paper.
>
> **Q2: Ground-truth answer**
>
> We want to clarify that GSPO needs the ground truth class labels y but does not have textual ground truth since it deals with LLM outputs. This is the "free-form target" challenge, as mentioned in our paper. Our solution involves constructing the textual ground truth that starts with "The final answer is str(y)" and only supervising the first sentence of the LLM's output with it. More details and strategies to complement this design are presented in lines 194-226 of our main paper.
>
> **Q3: Image-based text explanation**
>
> Sorry, we do not fully understand this question and are unsure what is meant by "social relationships used in baselines". Could you please elaborate further? We are more than happy to answer your question. We try to answer it below based on our current understanding.
>
> A social story is a textual paragraph depicting the social context of an image, while social relationships are the target categories, represented by a one-hot numerical vector without any textual explanations in previous social relation methods. Image-based text explanation is not the main focus of this paper. Instead, our paper aims to provide a zero-shot baseline for social relation recognition using foundation models. Building on this framework, the GSPO is proposed to automate the prompting process.

---

> > ### Author Response · Authors · 2024-08-12
> > **Looking forward to your feedback**
> >
> > Dear reviewer,
> >
> > Thank you for the comments on our paper. We have responded to your initial comments. We are looking forward to your feedback and will be happy to answer any further questions you may have.

---

> > > ### Comment · Reviewer_9pM3 · 2024-08-13
> > >
> > > I appreciate your response. However, I am still unclear about the differences between the proposed social story (in text form) and the graph-based social relationships used in baselines such as GR2N or TRGAT. Do you think text-based LLM prompts are superior to graph-based ones? For example, graphs can also be used as LLM prompts for reasoning [1]. If the social graphs used in the baselines also work well with LLMs, it seems difficult to argue that the social story contributes significantly.
> > >
> > > By the way, I also have read the other reviews that have been posted and their corresponding author responses. I will continue to read other reviews and the authors' responses and engage in further discussions with other reviewers until the end of the discussion period.
> > >
> > > [1] https://arxiv.org/pdf/2311.17076

---

> > > > ### Author Response · Authors · 2024-08-13
> > > >
> > > > Thank you for providing me the opportunity to clarify.
> > > >
> > > > To illustrate the difference, we first present the pipeline:
> > > >
> > > > Our pipeline is:
> > > > > Image -> [Foundation Models] -> Social Story (text form) -> [LLM] -> Prediction
> > > >
> > > > The pipeline of GR2N and TRGAT is:
> > > > > Image -> [Neural Net] -> Social Graph (feature representation) -> [Graph Neural Net] -> Prediction.
> > > >
> > > > The differences are significant: 1) Our social story is a **text-form image description** that can be input into LLMs. In contrast, the social graph in GR2N and TRGAT is a **feature representation** that cannot be input into LLMs. Therefore, the social graph in GR2N and TRGAT cannot work with LLMs. Our method is the first social relation recognition method that supports LLM reasoning. 2) The difference between text form and feature representation indicates a key difference in paradigms: our paradigm supports zero-shot social relation recognition with foundation models, achieving greater interpretability and generalizability, whereas existing social relation methods train a dedicated network end-to-end, limiting their interpretability and generalizability.
> > > >
> > > > As for the "graph" in [1], it has a different meaning than the "graph" in GR2N and TRGAT. The "graph" in [1] is also a text-form image description, not a feature representation. The pipeline in [1] is:
> > > > > Image -> [Foundation Models] -> Social Graph (text form).
> > > >
> > > > So, the "graph" in [1] is more like our text-based social story and shares the same motivation, but with a different way to achieve it. Therefore, the "graph" in GR2N and TRGAT cannot be used for LLM prompts.
> > > >
> > > > As we highlighted before, our main focus is not the specific instantiation of the social story; rather, it is to provide a framework for zero-shot social relation recognition using foundation models. The major technical contribution lies in our GSPO, which optimizes the SocialPrompt within our framework.
> > > >
> > > >
> > > > We hope this detailed explanation addresses your concerns. Please feel free to ask any further questions. We are more than willing to discuss and resolve any additional concerns you may have during the discussion period.
> > > >
> > > > [1] https://arxiv.org/pdf/2311.17076

---

> > > > > ### Author Response · Authors · 2024-08-14
> > > > > **Looking forward to your further response before the end of discussion period**
> > > > >
> > > > > Dear Reviewer,
> > > > >
> > > > > Thank you for your follow-up questions.
> > > > >
> > > > > As the discussion period is coming to an end, we would like to know if our responses have addressed your concerns. Please let us know if you have any further questions or need additional clarification, and we will be glad to assist.

---

### Author Response · Authors · 2024-08-11
**Thank you and we are looking forward to your post-rebuttal feedback!**

Dear AC and all reviewers:

Thanks again for all the insightful comments and advice, which helped us improve the paper's quality and clarity.

The discussion phase has been on for several days and we are still waiting for the post-rebuttal responses.

We would love to convince you of the merits of the paper. Please do not hesitate to let us know if there are any additional experiments or clarification that we can offer to make the paper better. We appreciate your comments and advice.

Best,

Authors

---

> ### Author Response · Authors · 2024-08-12
> **Could AC help to reach out reviewers?**
>
> Dear AC:
>
> Thank you for efficiently handling our draft.
>
> As the rebuttal phase is nearing its end, we haven't received any post-rebuttal feedback.
>
> Could the AC assist in reaching out to the reviewers for a response?
>
> We greatly appreciate all the efforts you've made!
>
>
>
>
> Thanks
>
> Authors

---

### Decision · Program_Chairs · 2024-09-25

**Decision:**

Accept (poster)

**Comment:**

The paper received 2 Borderline Reject, 1 Borderline Accept and 1 Accept ratings. While the authors should incorporate the reviewers' comments in their final version to elaborate the motivation of the storyline and the technical and experimental details, it is a solid contribution to the community. The AC reads the paper, the rebuttal, and the discussions and decides to accept it. Congratulations!